# The Effects of Stocking Density and Food Deprivation on Mucous Cells and Lysozyme Activity in the Skin and Gills of Silver Catfish

**DOI:** 10.3390/ani13223438

**Published:** 2023-11-07

**Authors:** Gabriela Pires Scherer, Isadora Maria Oliveira Zavaglia, Fernando Jonas Sutili, Hugo Napoleão Pereira Silva, Magale Dallaporta Furquim, Marcelo Leite da Veiga, Bernardo Baldisserotto, Sílvio Teixeira da Costa

**Affiliations:** 1Post-Graduate Program in Animal Science, Federal University of Santa Maria, Santa Maria 97105-900, Brazil; gabrielapiresscherer@gmail.com (G.P.S.); isa_zavaglia@hotmail.com (I.M.O.Z.); 2Eloaqua Consulting/Laboratório Scire, Frederico Westphalen 98400-000, Brazil; sutilifj@gmail.com; 3Post-Graduate Program in Animal Biodiversity, Federal University of Santa Maria, Santa Maria 97105-900, Brazil; hugofpa@hotmail.com; 4Department of Morphology, Federal University of Santa Maria, Santa Maria 97105-900, Brazil; magale.dfurquim@gmail.com (M.D.F.); marcelolveiga@gmail.com (M.L.d.V.); 5Department of Physiology and Pharmacology, Federal University of Santa Maria, Santa Maria 97105-900, Brazil

**Keywords:** mucous cells, gills, skin, stocking densities, food deprivation, morphometry, lysozyme

## Abstract

**Simple Summary:**

This study aimed to identify the regions of the skin of a South American catfish with a higher number of mucous cells. Additionally, the effects of stressful conditions that may occur in fish cultures, such as different stocking densities and food deprivation, on the number of mucous cells in the skin and gills and their impact on cutaneous mucous lysozyme activity were investigated. The sampled catfish were exposed to the conditions of high stocking density (32 kg/m^3^) and fed, high stocking density and fasted, low stocking density (2.5 kg/m^3^) and fed, and low stocking density and fasted. After 14 days, samples of cutaneous mucous and skin fragments from seven different points and the second left branchial arch were collected. The ventral point in front of the ventral fin was the portion with the highest number of mucous cells. The number of mucous cells in both the skin and gills varied based on the collection point and treatment applied. The highest lysozyme activity in the epidermal mucous was observed in fish exposed to high stocking density and fed. Stocking density and food deprivation create stressful conditions for this type of catfish, which modulate its mucosal response in the skin to each situation.

**Abstract:**

This study aimed to identify the regions of the body surface of silver catfish (*Rhamdia quelen*) with a higher population of mucous cells in the skin. Additionally, the effects of stressful conditions, such as different stocking densities and food deprivation, on the proliferative response of mucous cells in the skin and gill epithelium and their impact on cutaneous mucous lysozyme activity were investigated. Silver catfish were divided into four experimental groups: high stocking density (32 kg/m^3^) and fed (HSD-F), high stocking density and fasted (HSD-FS), low stocking density (2.5 kg/m^3^) and fed (LSD-F), and low stocking density and fasted (LSD-FS). Fish in the fed groups received commercial feed twice a day, amounting to 1% of the tank biomass. After a 14-day experimental period, the fish were anesthetized and euthanized. Samples of cutaneous mucous and skin fragments from seven different points and the second left branchial arch were collected. Histological slides of the skin and gills were stained with PAS + Alcian Blue at pH 2.5, and the epidermal mucous lysozyme activity was assessed using the turbidimetric method. The ventral point in front of the ventral fin was found to be the optimal location for collecting cutaneous epithelia due to its higher density of mucous cells. The population of mucous cells in both the skin and gills varied based on the collection point and treatment applied. The highest lysozyme activity in the epidermal mucous was observed in fish from the HSD-F group. Overall, these findings suggest that stocking density and food deprivation create stressful conditions for silver catfish, which modulate their mucosal response to each situation.

## 1. Introduction

The modern aquaculture industry demands alternative preventive practices that can help maintain high animal welfare and a healthy environment, resulting in better production and increased profits. A deeper understanding of the immune system of cultivated fish may assist in achieving these goals [1]. The immune defense system of fish consists of numerous distinct and interdependent immune components essential for defending against invading pathogens. Each element of the immune system holds its inherent protective value, and the ultimate combination of these components is likely associated with a satisfactory immune response [2].

The tegument or skin is the covering that not only separates and protects fish from their environment, but also provides the means through which most interactions with the outside world take place. The production of mucus helps protect these physical barriers, acting as a diffusion barrier and a lubricant whose function is to protect epithelial cells against infection, dehydration, and physical or chemical injury. Mucus also contains various bioactive components, such as immunoglobulin, complement C-reactive protein, lectins, lysozyme, proteolytic enzymes, alkaline phosphatase, and esterase, as well as antimicrobial peptides and hemolysin, which exhibit biostatic or biocidal activities [1,3]. It is a large and continuous organ, lining all body openings and also covering the fins. Besides serving as a mechanical barrier, it is a metabolically active tissue [4].

In routine aquaculture practices, fish are exposed to various stress conditions, both naturally occurring and artificially induced. Water quality factors, such as nitrite levels, temperature, dissolved oxygen, ammonia concentration, and salinity, as well as cultivation conditions like stocking density, water flow, and food availability, along with management procedures including biometrics and transportation, can collectively impact the physiological well-being of teleosts.

The immune system of fish comprises the systemic and mucosal immune systems. Since most infectious agents initiate the infection process on the mucosal surface, the mucosal immune system of fish serves as the first line of defense against encountering pathogens [1,3]. Stress has the potential to compromise the immune response, thus reducing the resilience and survival capacity of fish against pathogens [5,6]. Mucus, secreted on the skin surface by caliciform cells of the epidermal layer, presents a specialized composition that serves to prevent the adhesion and colonization of potentially harmful microorganisms and metazoan parasites. Furthermore, this secretion contains constituents linked to the metabolic activities of the immune defense mechanism. These include a variety of antibacterial agents, such as lysozyme, immunoglobulins, complement proteins, and antimicrobial polypeptides (AMPPs), which collectively exert inhibitory or lytic actions against a broad spectrum of pathogens [3,7]. Amongst these changes in the skin and gills of fish, which are associated with various stressful conditions, are the number and size of mucous cells located in the epidermis of the skin. These cells are responsible for producing the mucus that covers the fish’s body [2].

The silver catfish (*Rhamdia quelen*) stands out as a benthic species suitable for the South American consumer market due to its rapid growth rate, high carcass yield, and convenient reproductive management in subtropical climates [8]. It has undergone extensive cultivation and has even served as a model for refining management practices for various fish in this family [9]. Previous research has indicated the impact of population density on its growth [10,11].

The concentrations of mucous cells in different portions on the body surface have been examined in gilthead seabream (*Sparus aurata*) [12], brown trout (*Salmo trutta*), Artic charr (*Salvelinus alpinus*) [13], benthic striped catfish (*Pangasius hypophthalmus*), and Philippine catfish (*Clarias batrachus*), but not in silver catfish. A detailed analysis of the impact of hypoxia stress on cutaneous and branchial histology has been conducted in sea bass (*Dicentrarchus labrax*) [14] by observing three locations above and three locations below the lateral line, but an examination regarding the effects of stocking density and fasting on mucous cells on the body surface has not been analyzed yet.

Therefore, the present study aims to determine the ideal cutaneous location for histological analyses in silver catfish, considering cellularity and increased mucus production. Additionally, this study aims to examine the histological responses of the skin and gill tissues to potential stressors linked to variations in stocking density and food deprivation.

## 2. Materials and Methods

### 2.1. The Experiment and Animals

The fish (190.0 ± 5.0 g and 25.0 ± 1.0 cm) used in this study were bought from a local fish farm, transported to the laboratory, and acclimated for 10 days in eight 250 L tanks with continuous aeration and water renewal (10 L/min) under a photoperiod of 14 L:10 D. After this period, five animals were collected from different tanks, anesthetized with eugenol (50 mg/L) for 3 min [15], and euthanized by means of spinal cord sectioning for sample collection (see Section 2.3). This group was characterized as the basal group (stocking density of 10.8 kg/m^3^) of this experiment. The remaining fish were randomly redistributed in tanks with the water levels adjusted to 200 L, along with continuous aeration and water renewal (10 L/min). Silver catfish were assigned to four treatments (two replicates each): high stocking density (37 fish/tank 32 kg/m^3^) and fed (HSD-F); high stocking density and fasted (HSD-FS); low stocking density (3 fish/tank, 2.5 kg/m^3^) and fed (LSD-F); and low stocking and fasted (LSD-FS). Low stocking density and fasting are stressful for silver catfish [9,16,17]. The animals in the fed groups were fed twice a day with 1% of the tank biomass, at 8 am and 5 pm, using a commercial feed containing 42% crude protein, while those in the fasted groups were not fed during the 14 days of the experiment. This experiment was approved by the Ethics Committee on Animal Use of the Federal University of Santa Maria (number 1604060319).

### 2.2. Water Quality Parameters

The tanks were siphoned twice a day to remove feces and feed residues that had settled at the bottom of the tank, prior to feeding at 8:00 am and 5:00 pm. Water parameters were measured daily. Temperature (21.4 ± 0.7 °C) and dissolved oxygen (6.4 ± 0.2 mg/L) were determined using a YSI oxygen meter (Y5512, Yellow Springs, OH, USA), and pH (5.8 ± 0.04) was measured using a DMPH-2 pH meter (Digimed, São Paulo, SP, Brazil). Also measured were total ammonia (0.34 ± 0.26 mg/L) [18], alkalinity (36.0 ± 3.3 mg CaCO_3_/L), hardness (50.0 ± 0.0 mg CaCO_3_/L), and nitrite (0.15 ± 0.02 mg/L) [19]. These values are within the appropriate range for silver catfish [8].

### 2.3. Collection of Biological Material

After 14 days, 6 fish from each of the LSD groups and 8 fish from each of the HSD groups were anesthetized and euthanized as described in the previous section. The fish were considered experimental units as in a previous study dealing with the same conditions [17]. Skin mucus samples were collected according to established protocols [20,21]. The collected mucus was stored in 1.5 mL Eppendorf© tubes maintained at a refrigerated temperature of −80 °C for subsequent lysozyme activity analysis. Additionally, the second left gill arch and seven skin fragments taken from different points on the left side of the fish’s body surface (Figure 1) were collected. Samples were individually placed in 15 mL tubes containing 10% formaldehyde solution to facilitate subsequent routine histological processing. Upon paraffin embedding, 5 µm thick sections of both skin and gill tissues were subjected to PAS and Alcian Blue staining methods to analyze mucus-secreting cells [3,4,22]. According to [22], images of various regions were observed under an optical microscope, where an analysis of mucus-secreting cells was conducted in 15 to 25 fields for each section of the skin and gills of each sampled fish.

### 2.4. Biological Material Analysis

The slides were observed and photomicrographed with the aid of an Axio Scope A1 model microscope in a digital image capture system using an attached Axiocam 105 color camera (ZEISS^®^, Jena, Germany). After evaluating the percentage of secretory cells in relation to the epithelium using the ImageJ software (1.54d), a correction was made using the formula: (comparative area X% area/total area of the photo).

The mucus was blended with phosphate-buffered saline (PBS—0.01 M, pH 7.2) at a 1:1 ratio. The resulting samples underwent centrifugation at 7000× *g* for 10 min, with the ensuing supernatant carefully separated and preserved. The enzymatic activity of lysozyme in the fish mucus was determined using a methodology previously described [23]. Briefly, the mucus samples (10 µL) were mixed with a suspension of *Micrococcus lysodeikticus* (M3770, Sigma, St. Louis, MO, USA) in PBS (0.2 g/L, pH 6.2, an optimal pH for lysozyme activity in silver catfish [24]). The mixture was deposited in 96-well flat-bottom plates, and subsequent absorbance readings were taken at 450 nm utilizing a Synergy H1 Multi-Mode microplate reader. Lysozyme activity (units/mL) was calculated using the following formula:$$\mathrm{Lysozyme}\;\mathrm{activity}=[\frac{∆\;absorbance\frac{\left(4-1\;min\right)}{3}}{0.001}]\times 100$$

Egg albumen lysozyme (L6876, Sigma, St. Louis, MO, USA) was used as a standard, and a unit of lysozyme activity was defined as the amount of enzyme that produces a decrease in absorbance of 0.001/min.

### 2.5. Statistical Analysis

The normality of data distribution and the homogeneity of variances were assessed using Bartlett and Levene’s tests, respectively. However, the data did not conform to a normal distribution nor exhibited homoscedasticity. To compare the number of mucus-secreting cells across various points on the fish’s body surface in the basal group, Kruskal–Wallis one-way analysis of variance (ANOVA) on ranks was employed. Additionally, the effects of stocking density and feeding on each point on the fish’s body surface were subjected to comparison using the Kruskal–Wallis/Scheirer–Ray–Hare test, followed by the Nemenyi test. The minimum level of significance was established at 95% confidence (*p* < 0.05).

## 3. Results

### 3.1. Skin Analysis in the Basal Group

Fish from the basal group presented a greater number of cutaneous secretory cells in the DCDF, DDF, LDSB, and VVF than in the VCVF and LVLB (Figure 2A and Figure 3). On the other hand, the subcutaneous dermal thickness was the highest in the CO and DDF, followed by the LDSB, VVF, and LVLB. The lowest thickness was observed in the VCVF (Figure 2B).

### 3.2. Cutaneous Response to Stocking Density and Food Deprivation

Fasted fish maintained at both stocking densities had a lower number of goblet cells at all collection sites compared to fed fish. Only the DDF site in fish kept under the condition of HSD-F showed lower cellularity compared to fish kept under the condition of HSD-FS. The LSD-F fish presented a higher number of goblet cells at all collection sites compared to HSD-F fish, but in fasted silver catfish, the number of goblet cells was higher in those kept at low stocking density only in the CO, LDSB, VCVF, and VVF, being significantly lower in the DDF (Table 1). However, Table 3 demonstrates a higher activity of the lysozyme enzyme in fed fish with HSD. The morphometry of the conjunctival dermis of the subcutaneous tissue was not affected significantly by the treatments in any of the collected points.

### 3.3. Gill Response to Stocking Density and Food Deprivation

The number of branchial mucous cells was similar in the filaments and lamellae, irrespective of the treatments. Silver catfish maintained under the condition of low stocking density presented a lower number of mucous cells in the gill filaments and lamellae, but the feeding condition did not significantly affect the number of goblet cells in the gills (Table 2). Gill morphometric parameters such as height, thickness, and distance between lamellae were not significantly affected by the treatments.

**Table 2 animals-13-03438-t002:** Gill mucous cell count/linear mm in silver catfish (*Rhamdia quelen*) submitted to different stocking densities and feeding conditions.

	HSD	LSD
Gill filaments	77.59 ± 6.41	59.16 ± 7.27 *
Gill lamellae	76.24 ± 6.43	59.22 ± 7.29 *

Values are presented as mean ± SEM. n = 6–8. * Significantly different from HSD. ANOVA with Kruskal–Wallis test, *p* < 0.05.

**Table 3 animals-13-03438-t003:** Lysozyme activity in the epidermal mucus of silver catfish (*Rhamdia quelen*) submitted to different stocking densities and feeding conditions.

	HSD-F	HSD-FS	LSD-F	LSD-FS
Lysozyme activity	75.0 ± 12.2 ^c^	133.0 ± 22.7 ^a^	94.5 ± 13.3 ^b^	72.0 ± 10.2 ^c^

Values are presented as mean ± SEM (min–max). n = 6–8, ANOVA with Kruskal–Wallis test, *p* < 0.05. Different letters indicate significant difference between treatments. HSD-F: High stocking density (32 kg/m^3^) and fed; HSD-FS: High stocking density and fasted; LSD-F: Low stocking density (2.5 kg/m^3^) and fed; LSD-FS: Low stocking density and fasted. Between parentheses, numbers represent maximum and minimum values.

## 4. Discussion

The literature is still scarce in terms of information about how the cutaneous and subcutaneous layers vary at different points on the body surface of teleosts. The main constituents of the epidermis are epithelial cells. The surface of the epidermis, in general, is covered by a mosaic pavement of polygonal epithelial cells with varying dimensions and irregular distribution [25]. It is also understandable that the distribution of mucous cells in a small area of the epidermis is not always uniform [22]. However, it is recognized that cutaneous mucus is considered the first line of defense against infections through the skin epidermis [1]. As proposed for Atlantic salmon (*Salmo salar*) [26], quantification of cutaneous goblet cells in silver catfish was observed at three collection points above the left lateral line and three collection points below the lateral line, while in striped catfish and Philippine catfish, the observed regions included barbels and dorsal and ventral parts of the head and the abdomen [22]. Therefore, adding to the proposed model for Atlantic salmon [26], an additional collection point was chosen that was located caudally to the operculum, which shows a similar cellularity to the other body regions observed in silver catfish. It is possible to observe a greater cutaneous cellularity in silver catfish in the dorsal anteroposterior ventral direction, starting from the VVF and passing through the DDF and LDSB to the DCDF. Presumably, this is because of the benthic characteristic of the species due to this body region having greater contact with the solid substrates of riverbeds, lakes, and reservoirs and following the hydrodynamic flow during the movement of silver catfish in the watercourse.

Larger cell sizes were also observed in the dorsal region of gilthead seabream [12], but the highest concentrations of mucosal cells in brown trout and Arctic char were observed in the anterior body regions, with a lower number of mucosal cells in the fins [13]. The highest number of mucosal cells in striped catfish was found in the dorsal region of the tail, followed by the barbels in the head region and the head itself. A lower count of mucosal cells was observed in the abdomen. However, an inverse pattern was found in Philippine catfish, where the highest number of mucosal cells was found in the head region and the lowest in the tail [22].

In the ventral region of gilthead seabream, the epidermis is thicker compared to the dorsal region [4], with a greater number of mucous cells in the anterior region. Silver catfish also show thicker epidermis in the anterior region, but dorsally and not ventrally as in gilthead seabream [12]. Silver catfish is a benthonic and not very active species, with a preference for sheltered areas [27]. Despite this preference for sheltered areas, the location of mucous cells and the subcutaneous dermal thickness in the skin of silver catfish are apparently related to the laminar flow of water, as observed by [13] for brown trout and Artic charr.

The body surfaces of multicellular organisms are defended by epithelia, which provide a physical barrier between the internal environment and the external world. The skin is a structure that covers the body and protects it not only from the entry of pathogens or allergens but also from the leakage of water, solutes, or nutrients [1]. It is unequivocal that fish skin plays an important role as the first protective barrier against many invading agents. Furthermore, the homeostasis of mucus physicochemical factors is very important in preventing the potential invasion and/or adhesion of pathogens to mucosal surfaces [28]. Often, small nonspecific changes in fish skin morphology, mainly its color, can be indicative of a stressful condition or potentially dangerous circumstance. However, it has been shown that plasma cortisol changes as determined by stress are not related to the physiological mechanism of stress displayed in the skin of fish [2]. Food deprivation and low stocking densities have been described as stressful for silver catfish because both situations increase plasma cortisol and alter some metabolic parameters [9,11,16,17], and this agrees with the overall higher number of mucous cells in the skin of the LSD-FS fish. In agreement with our observations, other stressful situations such as a low oxygen level (3.6 mg/L) for 9 weeks and a high concentration of nitrate (700 mg/L) for 48 h increased the number of mucous cells per skin area in sea bass, but the diameter of the skin mucous cells and the skin structure were not altered by these factors [14].

Furthermore, the continuous secretion and elimination of mucus produced by cutaneous mucous cells are associated with many other substances, such as immunoglobulins, lectin, lysozyme, and complement, which protect fish against various infections [14]. The cutaneous mucus of fish subjected to the HSD-FS treatment showed greater lysozyme activity than those in the LSD-FS and HSD-F groups.

In the branchial filaments and lamellae of silver catfish, the response of mucous cells to stocking density was different from the skin, with a lower number observed in fish exposed to the lower stocking density and without a significant effect of fasting. The effect on gill mucous cells related to stocking density in silver catfish agrees with the lower number of mucous cells in the gills of Nile tilapia (*Oreochromis niloticus*) exposed to heat and cold shocks [29]. Is it possible that an increase in stocking density promotes an increase in metabolic waste and the microbiota of these fish in the experimental environment, driving them to a stronger response of their mucosal cells in the gills, thereby enhancing the microclimate in the filament and lamellae environment. Previous studies have shown that many biotic adversities, as well as abiotic environmental factors, such as increased ammonia concentration in water, acidification, organically fertilized water, and heavy metals, can affect the morphology and structure of fish skin, especially the distribution of mucous skin cells [2]. However, other types and duration of stress may alter the number of mucous cells in the gills, for example, amoebic gill disease in Atlantic salmon [30] and 1 day (but no significant difference after 15 days) of hypo- and hyperosmotic stress in *Hoplias malabaricus* [31].

## 5. Conclusions

The dorsal portion is the best area of silver catfish’s body to assess skin mucosa due to its higher number of mucous cells and also because it is a region that is more susceptible to the impact of environmental stress-related events or situations. Furthermore, the response of mucous cells appears to be associated with the body’s location, as stress response manifests with different patterns on the skin when compared to the gills.

## Figures and Tables

**Figure 1 animals-13-03438-f001:**
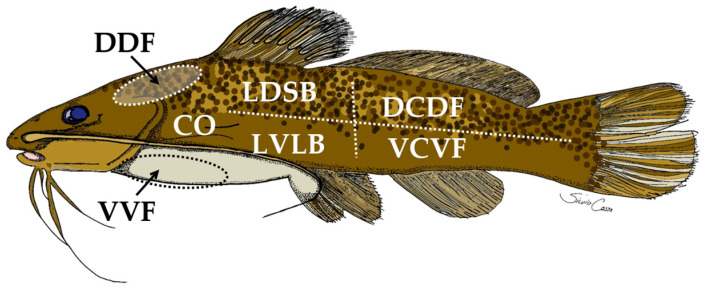
Representation of collection sites for obtaining skin samples from silver catfish (*Rhamdia quelen*). DDF—Dorsofrontal to dorsal fin; CO—Caudal to operculum; VVF—Ventrofrontal to ventral fin; LVLB—Left ventrolateral to lateral band; LDSB—Left dorsolateral to side band; DCDF—Dorsum caudal to dorsal fin; VCVF—Ventrocaudal to ventral fin.

**Figure 2 animals-13-03438-f002:**
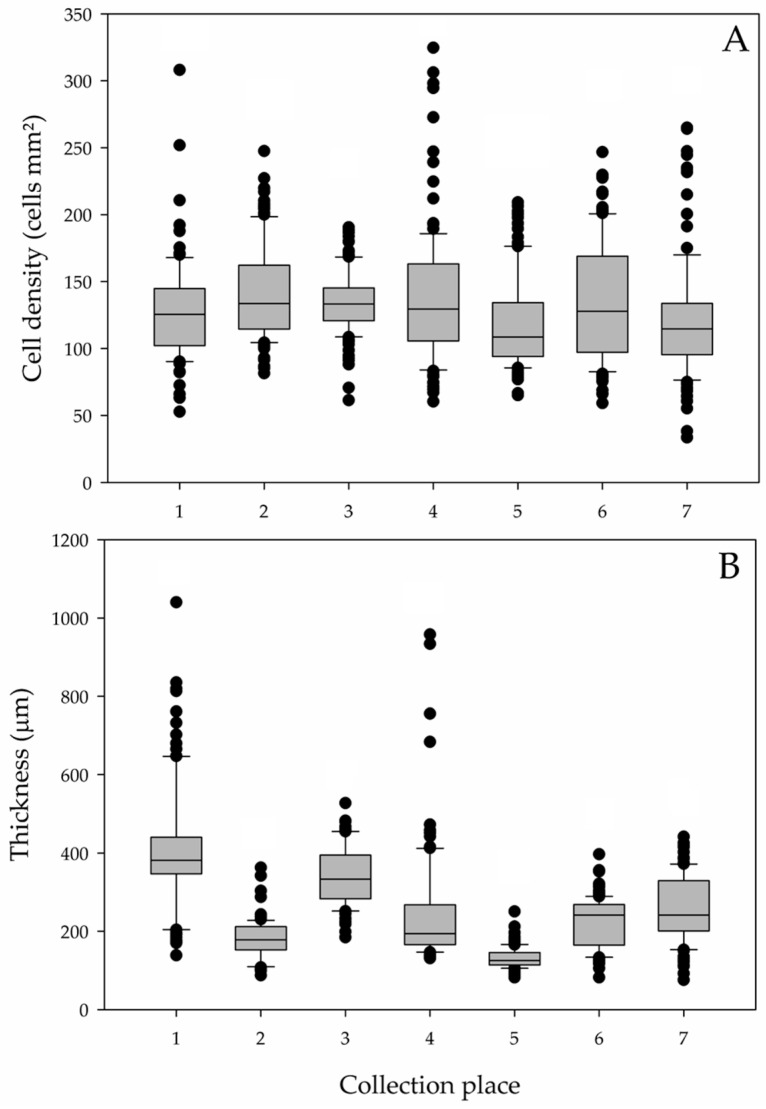
Quantitative parameters from the basal group. (**A**) Density of goblet cells. (**B**) Subcutaneous dermal thickness. The numbers represent the collection sites of the skin samples from silver catfish (*Rhamdia quelen*). 1. CO—Caudal to operculum; 2. DCDF—Dorsum caudal to dorsal fin; 3. DDF—Dorsofrontal to dorsal fin; 4. LDSB—Left dorsolateral to side band; 5. VCVF—Ventrocaudal to ventral fin; 6. VVF—Ventrofrontal to ventral fin; 7. LVLB—Left ventrolateral to lateral band. N = 6–8, Kruskal–Wallis one way analysis of variance (ANOVA) on ranks (*p* ≤ 0.001).

**Figure 3 animals-13-03438-f003:**
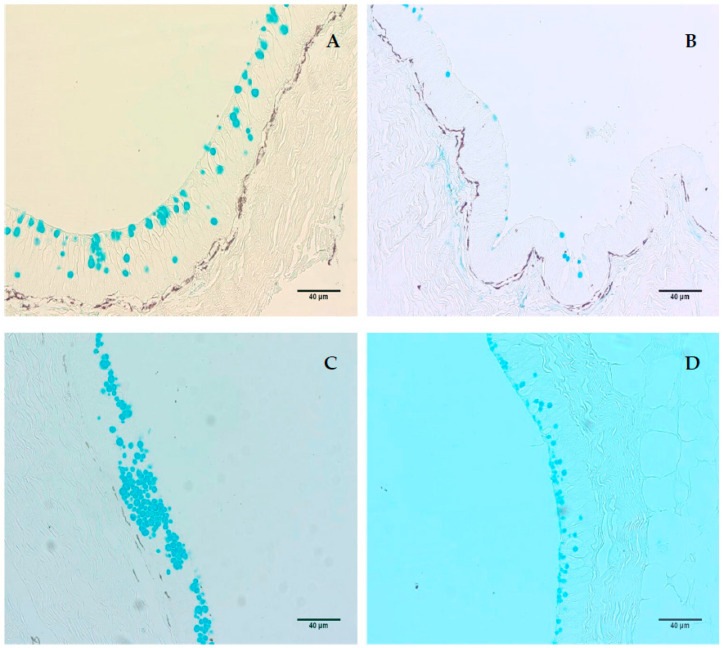
Cutaneous histology of silver catfish (*Rhamdia quelen*) under different stocking densities and feeding conditions. Photomicrographs of Alcian Blue-stained mucosal cells collected from the CO. (**A**) HSD-F: high stocking density (32 kg/m^3^) and fed; (**B**) HSD-FS: high stocking density and fasted; (**C**) LSD-F: low stoking density (2.5 kg/m^3^) and fed; and (**D**) LSD-FS: low stocking density and fasted (10× photomicrographs and 40 µm scale bar).

**Table 1 animals-13-03438-t001:** Score of cutaneous mucous cells in relation to the total area of the epithelium in different collection sites of silver catfish (*Rhamdia quelen*) submitted to different stocking densities and feeding conditions.

Collection Site		Fed		Fasted	
		LSD-F	HSD-F	LSD-FS	HSD-FS
CO	median	126.235 ^a^	102.423 ^b^	77.096 ^c^	56.687 ^d^
	25%	100.129	76.899	62.454	35.399
	75%	155.660	128.913	102.069	79.290
DCDF	median	148.339 ^a^	120.769 ^b^	88.611 ^c^	88.326 ^c^
	25%	98.722	100.862	69.422	64.780
	75%	179.823	154.366	121.049	120.124
DDF	median	111.610 ^a^	76.647 ^c^	79.654 ^c^	83.775 ^b^
	25%	88.865	59.114	65.486	57.406
	75%	132.596	102.554	98.327	99.057
LDSB	median	129.411 ^a^	86.423 ^c^	94.477 ^b^	70.143 ^d^
	25%	110.458	68.668	65.286	45.093
	75%	153.985	101.965	124.256	96.803
VCVF	median	137.414 ^a^	100.690 ^c^	105.819 ^b^	91.398 ^d^
	25%	117.477	87.176	81.955	68.303
	75%	156.255	121.582	126.348	117.334
VVF	median	161.006 ^a^	126.023 ^b^	115.069 ^c^	72.141 ^d^
	25%	137.196	96.673	94.864	53.418
	75%	188.434	157.191	162.466	105.813
LVLB	median	111.644 ^a^	91.869 ^b^	88.724 ^c^	76.797 ^c^
	25%	88.502	78.320	42.892	40.609
	75%	154.810	112.583	111.305	121.051

n = 6–8, ANOVA with Kruskal–Wallis test, *p* < 0.05. Different letters within the rows indicate significant difference between treatments. HSD-F: High stocking density (32 kg/m^3^) and fed; HSD-FS: High stocking density and fasted; LSD-F: Low stocking density (2.5 kg/m^3^) and fed; LSD-FS: Low stocking density and fasted. DDF—Dorsofrontal to dorsal fin; CO—Caudal to operculum; VVF—Ventrofrontal to ventral fin; LVLB—Left ventrolateral to lateral band; LDSB—Left dorsolateral to side band; DCDF—Dorsum caudal to dorsal fin; VCVF—Ventrocaudal to ventral fin.

## Data Availability

Data are available upon request to the authors.

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
