# Peer review of "The Effects of Stocking Density and Food Deprivation on Mucous Cells and Lysozyme Activity in the Skin and Gills of Silver Catfish"

_animals, 2023, doi:10.3390/ani13223438_

Round 1

Reviewer 1 Report

Comments and Suggestions for Authors

The paper context is clearly explained, however, some issues in method and analysis should be also addressed prior to publication.

Title: The title should be changed to something like: “The effect of stocking density and food deprivation on mucus cells and lysozyme activity in the skin and gills of silver catfish”

Methods and results

-       Generally, the work is simple and only few parameters that were assayed. More parameter should be assayed e.g. investigating inflammatory and suppressive modulation!

-       Differentiate between skin and gills in term of methods results in writing and figures including histological investigations and figures. The results of Masson-Goldner trichrome technique is missing and should be provided and completed.

Author Response

Reviewer 1

Title:

“The title should be changed to something like: “The effect of stocking density and food deprivation on mucous cells and lysozyme activity in the skin and gills of silver catfish.”

Understanding that the title of the manuscript can be improved, reviewer 1's suggestion is welcome.

“The effect of storage density and food deprivation on mucous cells and observed gill lysozyme activity of silver catfish – Rhamdia quelen.”

Methods and results:

Jundiá is a species native to Latin America with little published information when compared to other teleost species such as tilapia, salmon and sea bream. Even though it seems to be something commonplace, it is the first time that a morphological/morphometric investigation has been carried out on the skin of the Rhamdia quelen species. Thus, information regarding the body site with the highest density of mucous cells, as well as its response to stimuli considered disturbing (stress), was also never evaluated.

The investigation of inflammatory and suppressive modulation in conditions similar to those to which fish were subjected is, certainly, an excellent subject for new scientific investigation in our line of research, in the group and we thank you in advance for the pertinent suggestion.

  Regarding the presentation of images prepared with the Masson-Goldner trichrome technique, this technique widely used to qualify and identify types of collagen, differentiate them from muscle tissue and does not allow us to differentiate the type of secretion produced by mucous cells which, according to Esteban, M.A. (2012), PAS and alcian-blue stainings allow.

Reviewer 2 Report

Comments and Suggestions for Authors

Why do you to want to determine the ideal cutaneous location? What is the importance of skin and gill resonses to aquaculture environmental stressors? What do you want to obtain from performing stocking density and feed deprivation? In fact, catfish are originally a type of scaleless fish, with more abundant skin mucus than scaled fish. The rationality and necessity of the study are not described clearly. 

Two feeding regimes (feed deprivation and normal feeding) were adopted in the study. What is your purpose to apply feed deprivation treatment in the case of high and low stocking sensity? The feed deprivation duration is as long as 14 days. 

Please give an explanation about your choice. 

Usually, three replicates are the minimum requirement, but two replicate tanks were used for each treatment in this study. Why? 

The fish number for each tank under high and low stocking density needs to be stated. 

The rearing aquaculture system must be described clearly.

Two-way ANOVA  is prefered to one-way ANOVA.

What is the use of Figure 2?  What  is the relationship between this figure and the experimental design.

Table 1 did not present institutive results on both normal feeding and feed deprivation as well as high and low stocking density.

Figure 3 presented inconsistent image colors.

There is only lysozyme activity of epidermal mucus of silver catfish, insufficient immune indices to support the results. 

Author Response

Reviewer 2

Why do you wish to determine the ideal cutaneous location? What is the importance of cutaneous and branchial reactions to environmental stressors in aquaculture?

Answer: Initially, this information is unknown for silver catfish and other native species of Latin America. The presence, location, and distribution of mucous cells on the skin, as well as their potential response to stressful events, are also unheard of in these species. These cells, already described in marine species found in the Northern Hemisphere, produce secretions that may have various applications on the skin's surface, such as reducing friction, improving hydrodynamics in water movement, and secreting enzymes and peptides with antimicrobial action.

Therefore, as an additional way to assess impacts, the skin, which serves as a significant first barrier against external aggressors, can also be considered in terms of its morphological response in environmental assessment protocols. Thus, the choice of a cutaneous region that can express a better reactivity to stimuli measured by the quantification and qualification of mucous cells represents an important parameter for biological evaluation.

What do you hope to achieve by assessing animal density and food deprivation?

Answer: Stocking density and food deprivation represent just two of the handling adversities that fish can be subjected to in a fish culture. As biochemical and metabolic effects related to these factors have already been described in silver catfish, we investigated if the effects of these factors on the manifestation of cutaneous cellularity can reveal the tissue's susceptibility to this type of stress. As silver catfish is a scaleless fish and has this cutaneous secretion as a characteristic, could stress interfere with this physiological ability of the fish?

Two feeding regimes (food deprivation and normal feeding) were adopted in the study. What is your purpose in applying the food deprivation treatment in the case of high and low stocking density? The duration of food deprivation is up to 14 days.

Answer: Food deprivation (in this study we tested an extreme situation of food deprivation) and stocking density, as mentioned earlier, are two examples of stress events to which fish are subjected under cultivation conditions. The goal is to test if there is an interaction of these stress events in the cutaneous response of silver catfish and what this response would be like in 14 days.

Usually, three replicates are the minimum requirement, but two replicate tanks were used for each treatment in this study. Why?

Answer: Yes, experimentation conducted in triplicates provides excellent result reliability. However, the conditions available for this work did not allow us to carry out the study with three replicates. We recently published an article with some data of this experiment:

Silva, H.N.P. et al. Stress response of Rhamdia quelen to the interaction stocking density – Feeding regimen. General and Comparative Endocrinology, 2023, 335, 114228. https://doi.org/10.1016/j.ygcen.2023.114228.

The number of fish for each tank under high and low population density needs to be indicated.

Answer: we added the number of fish for each tank in each stocking density in lines 74 and 76.

The aquaculture rearing system needs to be described clearly.

Answer: After the initial collect, the fish were redistributed in 250 L capacity tanks in an open circuit, with continuous water renewal and aeration. We added these details in line 73.

Two-way ANOVA is preferable to one-way ANOVA.

Answer: the information in this section was incomplete. The ANOVA Kruskal-Wallis one-way analysis of variance on ranks was employed only to compare the number of mucus-secreting cells across various points on the fish's body surface in the basal group. We corrected this detail in line 146. The comparisons involving stocking density and fasting were done with the Kruskal-Wallis/Scheirer-Ray-Hare test, followed by the Nemenyi test, which is a “two-way analysis” for non-parametric data.

What is the utility of Figure 2? What is the relationship between this figure and the experimental design?

Answer: This figure aims to present  which region of the silver catfish's body surface best identifies the expression of mucous cell density in relation to the thickness of the lining epithelium for the basal group. Table 1 did not provide conclusive results for both normal feeding and food deprivation, as well as for high and low stocking density.

Figure 3 showed inconsistent image colors.

Answer: The image in Figure 3 explores the density of mucous cells in a skin sample collected in the caudal region adjacent to the gill operculum and demonstrates that in different treatments, there is a clear reflection of the stress of food deprivation in the manifestation of these cells on the cutaneous epithelium.

There is only lysozyme activity in the silver catfish's epidermal mucus, and the immune indices are insufficient to support the results.

Answer: The presence of lysozyme activity in the epidermal mucus of fish is a well-documented phenomenon in the existing scientific literature. Lysozyme is an enzyme that plays a crucial role in the innate immune system of many aquatic organisms, including fish. It serves as a first line of defense against bacterial infections by breaking down bacterial cell walls. In fish, lysozyme is commonly found in various mucosal secretions, including the skin mucus. The measurement of lysozyme activity in the epidermal mucus is an established method for assessing the immune response in fish, as it provides valuable information about their ability to resist bacterial infections and maintain mucosal homeostasis.

The presence of lysozyme in the epidermal mucus is consistent with the study's focus on the mucosal response of silver catfish to different stressors, such as stocking density and food deprivation. Regarding the sufficiency of immune indices to support the study's results, it's important to note that the study's aim was not to comprehensively assess all aspects of the immune response in silver catfish. Instead, the primary focus was on the proliferative response of mucus cells and the impact of stressors on epidermal mucus lysozyme activity.

Round 2

Reviewer 1 Report

Comments and Suggestions for Authors

I recommend publishing this version of the manuscript.

Author Response

Dear reviewer,
We would like to express our heartfelt gratitude for your invaluable contribution and the time you dedicated to reading and reflecting upon this material, which has undoubtedly enhanced the quality of our manuscript.
Unless you have any further remarks, I believe we have adequately addressed your feedback in "round 2." Nevertheless, we remain at your disposal to address any additional concerns you may have.
In our recent revisions, the additions and corrections are highlighted in yellow, and the final ones in gray.

Reviewer 2 Report

Comments and Suggestions for Authors

The title is still inaccurately described because mucous cells of skin and gill are expressed just not the number itself in regard to the result presentations of them in the text. Furthermore, only the lysozyme activity in gills was determined, but not for the skin.

The citation order of articles is incorrectly arranged.

Line 112-113: The statement of the sentence is hard to understand.

Line 185: What does the basal group mean for? The expression of the term should be consistent across the text.

When values in all tables and figures are shown as the means ± standard deviation, the number of replicates must be provided as the means ± SD (n = ? fish or ? tanks).

Table 3 only presented gill data of HSD and LSD. I strongly suggest that presenting the results in the form of HSD-F, HSD-FS, LSD-F, LSD-FS.

Comments on the Quality of English Language

no comments

Author Response

Hello!
The manuscript's title has already been modified as suggested by reviewer 1;

In lines 112 and 113, the sentence informs that, after the acclimatization period, five specimens chosen randomly were euthanized to obtain histological and morphometric information about the species without the interference of the treatments. These were therefore referred to as the baseline/control group;

The baseline group consists of animals that, after the acclimatization period and without interference in stocking density or food supply, provided histological and morphometric data for the study;

Table 3 does not show information regarding feeding or deprivation thereof because there were no differences between these treatments. Only stocking density affected the number of gill mucous cells.

In the attached manuscript, the modifications that were made to the manuscript are highlighted in yellow and gray. The final ones are highlighted in gray.

We hope we have addressed all your questions and remain available for any further information.

We are grateful for your willingness to observe, inform and allow us to improve our manuscript.
Att.
